# Investigating the chaotic dynamics produced by deep reinforcement learning controllers

## Abstract

In recent years, deep Reinforcement Learning (RL) has demonstrated remarkable performance in simulated control tasks however there have been significantly fewer applications to real-world problems. While there are several reasons for this dichotomy, one key limitation is a need for theoretical stability guarantees in real-world applications, a property which cannot be provided by Deep Neural Network controllers. In this work, we investigate the stability of trained RL policies for continuous control tasks and identify the types of dynamics produced by the Markov Decision Process (MDP). We find the solutions produced by this interaction are deterministically chaotic with small initial inaccuracies in sensor readings or actuator movements compounding over time producing significantly different long-term outcomes, despite intervention in intermediate steps. The presence of these chaotic dynamics in the MDP provides evidence that RL controllers produce unstable solutions, limiting their application to real-world problems.

## 1 Introduction

Modern control systems rely heavily on closed-loop controllers to regulate their state without human intervention. Through this feedback, the controller (policy) can continually monitor and adjust the control system (environment), guiding it towards a desired outcome as specified by a general reward function. In this framework, commonly represented as a Markov Decision Process (MDP) (Bellman, 1957), the policy selects actions based on the current state of the environment, allowing it to dynamically respond to any external disturbances. However, due to the indirect nature of this feedback, if the policy is not properly designed, the policy-environment interaction can produce oscillations in the system's state which compound to produce unstable dynamics. This can be extremely detrimental to real-world environments that operate with expensive equipment. Therefore, in addition to maximising long-term rewards, a policy should be optimised to produce stable solutions however, this is often overlooked when producing control policies.

One method which learns approximate solutions to the MDP which does not directly account for these unstable dynamics is deep Reinforcement Learning (RL) (Sutton & Barto, 1998; Mnih et al., 2013; 2015; Silver et al., 2016; 2017a;b; Lillicrap et al., 2019). In this framework, the policy is encoded by a Deep Neural Network (DNN) trained to take actions that maximise the feedback given by the reward function. However, when performing this optimisation little attention is given to the stability of these solutions instead focusing primarily on long-term reward maximisation. This produces controllers which are highly skilled at performing a specified task but are very sensitive to initial conditions. As a result, small initial changes to the environment can compound to produce significantly different long-term outcomes as shown in Figure 1. This sensitivity is a defining characteristic of a chaotic system (Lorenz, 1963; Devaney, 2003) and indicates the control policy is highly unstable. This poses a significant problem for the application of RL to real-world applications as these systems often require stability guarantees (Dulac-Arnold et al., 2019; 2021).

However, conducting stability analysis with deep RL controllers presents a significant challenge due to the complex non-linear nature of the DNN controllers. This makes it difficult to analyse and predict their behaviour using traditional stability analysis techniques as these are typically designed for linear controllers and simple control systems. However, quantifying the level of chaos produced by a fixed non-linear closed-loop system is a well studied problem within the context of dynamical systems (Liapunov, 1892).

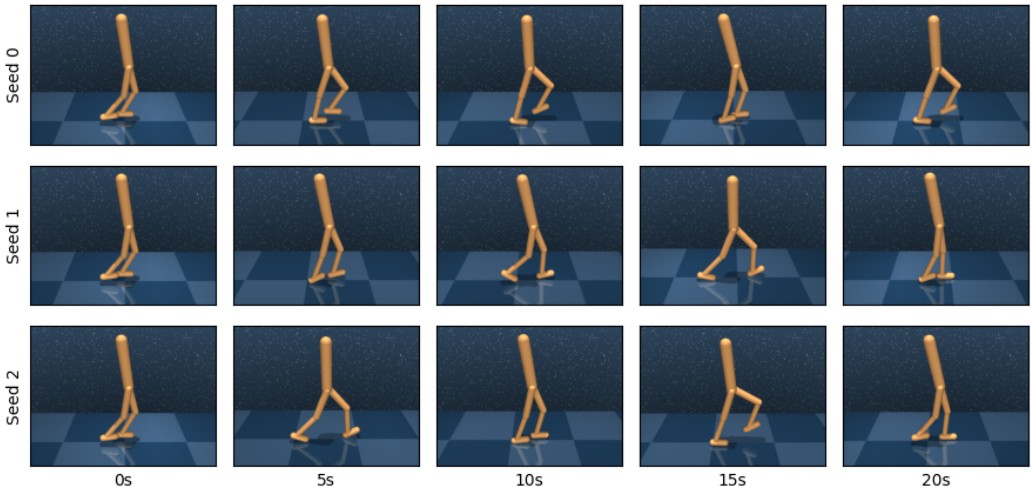

Figure 1: Long-term trajectories produced when one trained Soft Actor-Critic policy controls the *Walker Walk* Environment with different starting positions. Initially, each environment is separated by a distance of $10^{-6}$ units but significantly diverges after 5 seconds.

In this work, we use the mathematical analysis found in dynamical systems theory and chaos theory to quantify the stability of deep RL policies. We show the MDP framework can be redefined as a controllable dynamical system with the interaction between policy and controller being represented as a state trajectory. Using this representation, we analyse the stability of the interaction by measuring how sensitive trajectories are to initial conditions through the use of the Maximal Lyapunov Exponent (MLE) (Mawhin, 2005) and Sum of Lyapunov Exponents (SLE). From this, we establish that RL policies produce stable dynamics when controlling simple low-dimensional systems as small changes to the initial state do not produce significantly different long-term outcomes. However, as the complexity of the environment increases so too does the instability of the solutions.

Finally, we propose a method for reducing the unstable dynamics produced by the MDP by constraining all system states in the reward function. In general, the reward function in the MDP framework is designed to be as abstract as possible to allow for a wide range of potential solutions. However, this introduces an under-specification issue as only a subset of the system state is used when computing the reward. We show this ambiguity produces controllers which are highly sensitive to initial conditions and when properly constrained the instability reduces significantly. Therefore, properly constraining all system states in the reward function introduces a trade-off between stability and flexibility.

## 2 BACKGROUND

### 2.1 DYNAMICAL SYSTEMS

Dynamical systems are a general-purpose mathematical framework used to analyse the behaviour of complex, high-dimensional, non-linear systems over time. Each system is uniquely defined by a phase space $\mathbb{S}$, time domain $\mathbb{T}$ and update function $u : \mathbb{T} \times \mathbb{S} \to \mathbb{S}$. For systems with continuous states, $\mathbb{S}$ is a Euclidean space with each axis representing a different degree of freedom and the coordinates corresponding with a unique system configuration. Given an initial state $s_0 \in \mathbb{S}$, the update function for a continuous time dynamical system determines the state at time $t \in \mathbb{T}$ by the initial value problem $s_t = u(t, s_0)$. For a discrete time dynamical system, this update rule is recursively defined as $s_t = u(s_{t-1})$. Furthermore, the trajectory of an initial state $s_{t_0}$ is defined as the ordered set of states between times $t_0$ and $t_1$, i.e $\{s_t : t \in [t_0, t_1]\}$. Through this information, we can gain a greater understanding of how the system behaves over time as different trajectories reveal different system properties.

## 2.2 SYSTEM STABILITY

Within the domain of dynamical systems, stability is characterised by a tendency for trajectories to converge over time. Conversely, if any arbitrarily small perturbation to the state of the system produce significantly different long-term outcomes the system is said to be unstable and have a high dependence on initial conditions. While there are several methods for quantifying this stability one method which works well for complex non-linear systems is the spectrum of Lyapunov Exponents ($\lambda_i$) (Liapunov, 1892; Ruelle, 1979). These values represent the exponential rate of convergence/ divergence in each dimension of the phase space with negative values indicating convergence along an axis and positive values indicating divergence. In general, for a dynamical system with $N$ degrees of freedom there are $N$ Lyapunov exponents $\{\lambda_1, \lambda_2, ..., \lambda_n\}$ with $\lambda_i \geq \lambda_{i+1}$. Given a set of Lyapunov Exponents $\{\lambda_i : i \in [1, N]\}$ the average separation between two trajectories with initial distance $\epsilon_0$ is:

$$\epsilon_t = \sum_{i=1}^{N} e^{\lambda_i t} \tag{1}$$

In this equation, each Lyapunov exponent $\lambda_i$ represents the exponential growth rate in a specific direction in the phase space while the Sum of all Lyapunov Exponents (SLE) represents the average growth rate of an N-dimensional volume. Thus a negative SLE indicates stable dynamics as volumes in the phase space exponentially converge. Furthermore, a positive SLE is an indication of unstable dynamics as any small changes in the system state will produce exponentially diverging trajectories.

However, for large values of $t$ Equation 1 is dominated by the Maximal Lyapunov Exponent $\lambda_1$ (MLE) and the distance between two trajectories with initial separation $\epsilon$ approaches $\epsilon \times e^{\lambda_1 t}$ as $t \to \infty$. Thus, given two initial states $s_0$ and $\hat{s}_0$, with $|s_0 - \hat{s}_0| = \epsilon$, the MLE is found by the limit:

$$\lambda_1 = \lim_{t \to \infty} \lim_{\epsilon \to 0} \frac{1}{t} \ln \left( \frac{|s_t - \hat{s}_t|}{\epsilon} \right) \tag{2}$$

Values of $\lambda_1 < 0$ indicate the dynamical system is robust to small perturbations as similar states exponentially converge over time. Conversely, values of $\lambda_1 > 0$ suggest the system is unstable as small changes to the initial conditions result in vastly different trajectories. As a result, MLE can be used to quantify the stability of dynamical systems as negative values only occur in stable systems (Shao et al., 2016).

Finally, for a dynamical system with positive MLE and negative SLE the trajectories produced are said to be deterministically chaotic (Kalnay, 2003). This means that while there is a high sensitivity to initial conditions and exponential divergence the trajectories remain bounded by a chaotic attractor (Grebogi et al., 1987). As a result, once a trajectory has entered a chaotic attractor it becomes increasingly difficult to predict the long-term outcomes given an approximation of a state. Specifically, any trajectory prediction using an approximate state will on average decrease in accuracy by a factor of 10 every $\frac{\ln 10}{\lambda_1}$ seconds. This value is known as the Lyapunov Time and provides an upper bound on prediction accuracy given a state approximation.

## 3 REDEFINING THE MDP AS A DYNAMICAL SYSTEM

In order to use the MLE to determine the stability of RL controllers, the underlying MDP must be represented as a dynamical system. By making this redefinition the solutions produced by the policy-environment interactions can be viewed as state trajectories in the dynamical system and analysed accordingly. For this, an appropriate phase space and transition function need to be defined which can suitably represent the evolution of the control system over time. As chaos theory requires the phase space to be a continuous Euclidean space it is appropriate to use the environment state space as the phase space. It is worth noting that, for some environments, the observation space could be used for this however this requires each observation to be unique, a property which is not often given.

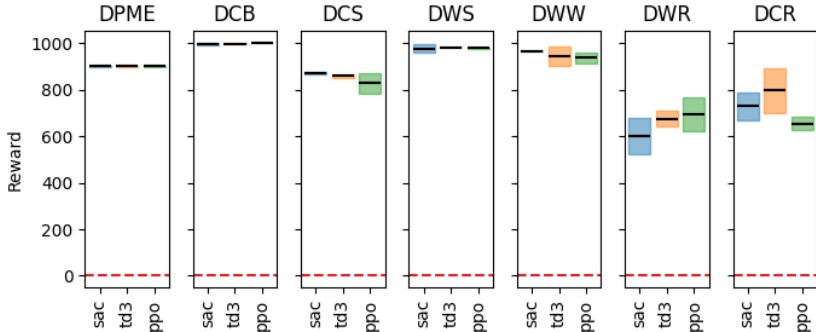

Figure 2: Boxplot of average total episode reward for the *Pointmass Easy* (PME), *Cartpole Balance* (CB), *Cartpole Swingup* (CS), *Walker Stand* (SW), *Walker Walk* (WW), *Walker Run* (WR) and *Cheetah Run* (CR) environments controlled by a trained instance of SAC, TD3 and PPO. Each policy-environment combination is independently trained with 10 random seeds and the average episode reward is reported over 80 evaluation episodes with a fixed length of 1000.

The transition function for this dynamical system must therefore map system states onto itself and be capable of determining the evolution of the system for all possible states. This can be produced by composing the policy, $\pi : \mathbb{S} \times \phi \rightarrow \mathbb{A}$, and environment transition function, $f : \mathbb{S} \times \mathbb{A} \rightarrow \mathbb{S}$, giving $s_{t+1} = u(s; \phi) = f(s_t, \pi(s_t; \phi))$. While this transition function removes the dependence on the action space, $\mathbb{A}$, and maps states to states it is now dependent on the policy parameters $\phi$. As such, any changes to $\phi$ produce a new dynamical system however for a fixed policy the dynamical systems remain fixed. Therefore, for a fixed control policy and deterministic environment, the MDP can be represented as a dynamical system and the stability of this interaction can be determined by estimating the MLE and SLE.

## 4 IDENTIFYING CHAOTIC DYNAMICS

By redefining the RL control loop as a controllable dynamical system we can use the trajectories produced by the update rule to determine if the policy-environment interaction is stable, unstable or chaotic. To closely match real-world applications we use tasks sampled from the DeepMind Control Suite (Tassa et al., 2018) as this provides a range of deterministic continuous control environments with varying complexity. Furthermore, as the state space for each of these tasks is a vector of joint angles, joint angle velocities or 3d coordinates it can appropriately be used as the phase space for a dynamical system. For each control task, a Stable Baselines 3 (Raffin et al., 2021) instance of Soft Actor-Critic (SAC) (Haarnoja et al., 2018), Twin Delayed Deep Deterministic Policy Gradients (TD3) (Fujimoto et al., 2018) and Proximal Policy Optimisation (PPO) (Schulman et al., 2017) in order to identify the types of dynamics produced by off-policy and on-policy deep RL actor-critic methods. Each model is independently trained 10 times for 5 Million (SAC, TD3) or 10 Million (PPO) environment steps and the final reward for each model type is reported in Figure 2. This established similar performance across all tasks however this does not speak to the types of dynamics produced by each policy as well as the stability of these solutions.

To identify if the dynamics produced by each policy and environment is stable, unstable or chaotic we need to determine the Sum of Lyapunov Exponents and the Maximal Lyapunov Exponent. This can be achieved by estimating the full spectrum of Lyapunov Exponents using the approach outlined by Benettin et al. (1980a;b). This method estimates the spectrum of representing small perturbations a set of orthogonal vectors in the phase space then iteratively updating these using the dynamical system transition function. Moreover, to avoid all vectors collapsing to the direction of maximal growth they are periodically Gram-Schmidt orthonormalized so they each maintain a unique direction. The Spectrum of Lyapunov Exponents are then determined as the average log convergence/ divergence of the perturbation vectors. Performing this orthonormalization allows for the detection of both positive and negative Lyapunov exponents up to the dimension of the phase space.

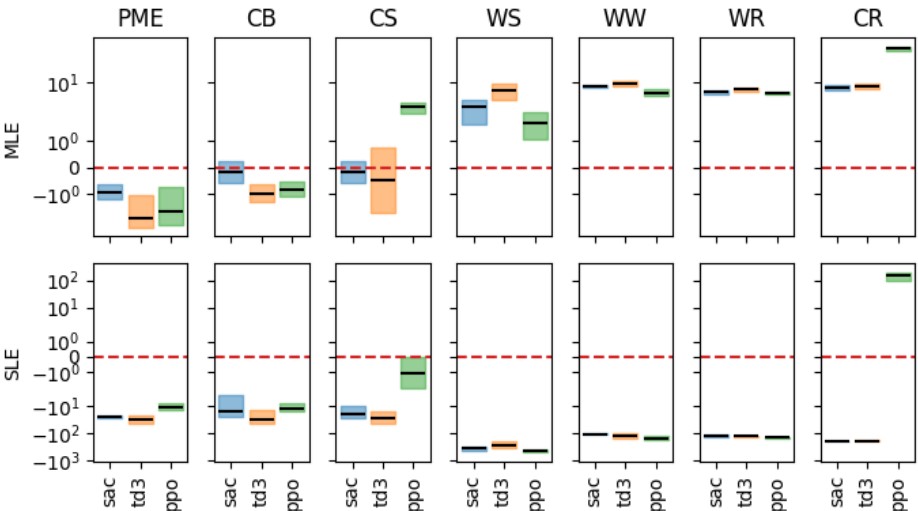

Figure 3: Estimated Maximal Lyapunov Exponent (MLE) and Sum of Lyapunov Exponents (SLE) for the *Pointmass Easy* (PME), *Cartpole Balance* (CB), *Cartpole Swingup* (CS), *Walker Stand* (SW), *Walker Walk* (WW), *Walker Run* (WR) and *Cheetah Run* (CR) environments controlled by a trained instance of SAC, TD3 and PPO. Each policy-environment combination is independently trained with 10 random seeds and the average MLE & SLE are calculated using 8 initial states.

Using this method we are able to identify the full spectrum of Lyapunov Exponent for each policy environment-environment interaction and derive the SLE and MLE. To provide an accurate estimate of these values we calculate the spectrum for 8 unique initial states taken from sample trajectories with a minimum length of 200 as this allows for the trajectory to reach any potential chaotic attractors. A perturbation vector is then initialised for each dimension of the phase space at a distance of $10^{-5}$ from the sample state. The spectrum is calculated over 10 thousand environment steps with orthonormalization every 10 steps and Figure 3 provides the MLE and SLE for each policy-environment interaction. From this, we can establish stable dynamics occur in the *Pointmass* and *Cartpole* environments as they have negative SLE and negative MLE. This result is consistent with the reward function for each of these tasks as they provide high rewards for maintaining the robot in a specified state. Moreover, as the reward function is dependent on each dimension of the state, the RL policy is trained to minimise any variation in all dimensions. However, the notable exception for these simple environments occurs when a trained *PPO* agent controls the *Cartpole Swingup* task as this produces negative SLE and positive MLE. As such, the interaction can be classed as chaotic with trajectories which are highly sensitive to initial conditions but bounded by a chaotic attractor. This difference in stability is likely due to the small perturbations causing the system to move outside the on-policy training distribution.

Furthermore, chaotic dynamics also occurs in complex high-dimensional environments (*Walker*, *Cheetah*) however, this is now consistent across all model types. As a result, despite attaining high rewards, these controllers cannot account for small perturbations to the system state as this produces significantly different long-term outcomes. Henceforth, the specific trajectory of a state cannot be predicted using an approximation of the state and true transition function as any small differences in the state estimation cause exponential diverging dynamics up to the size of the chaotic attractor. Moreover, despite varying levels of chaos found across environments, the level of chaos produced by each policy is relatively consistent. This provides an indication the chaotic dynamics are produced by the environment and is not accounted for by the control policy. Furthermore, Figure 4 shows the SLE and MLE for each policy-environment interaction during training and further shows stability is not directly optimised during the training process. The notable exception occurs when *PPO* controls the *Cheetah Run* environment as the MLE and SLE both increase during training and converge to positive values. As a result, the trajectories for this environment have a tendency to diverge indefinitely are are not bounded by a chaotic attractor. This demonstrates that the trajectories produced by the MDP can be stable, chaotic or unstable despite attaining similar rewards.

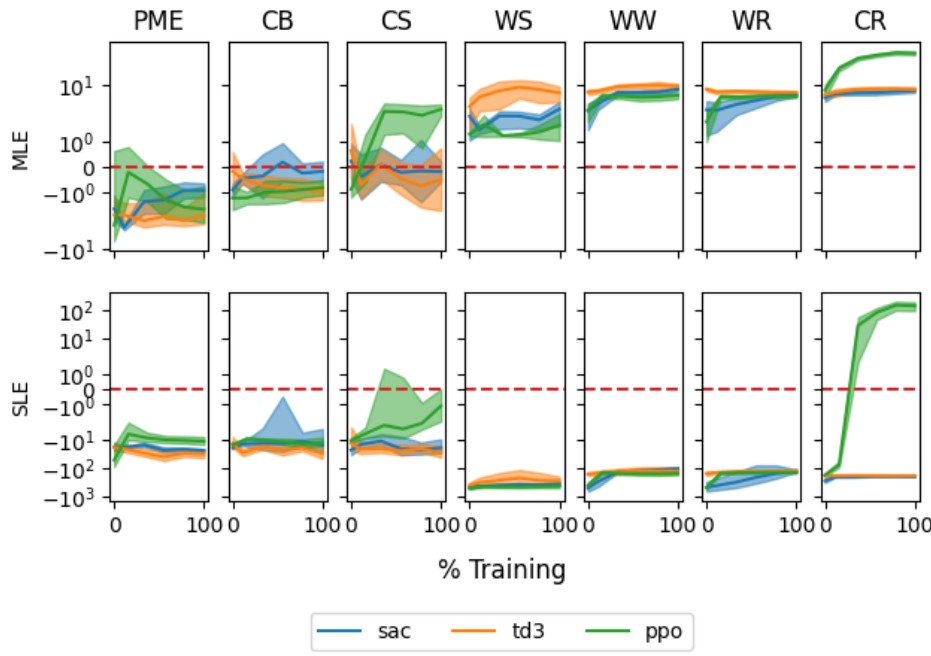

Figure 4: Average environment MLE and SLE during training for the *Pointmass Easy* (PME), *Cartpole Balance* (CB), *Cartpole Swingup* (CS), *Walker Stand* (SW), *Walker Walk* (WW), *Walker Run* (WR) and *Cheetah Run* (CR) when controlled by SAC, TD3 and PPO. Each policy-environment combination is independently trained with 10 random seeds and the average metrics are calculated using 8 initial states.

## 5 REWARD IN A CHAOTIC ATTRACTOR

Having established the presence of unstable and chaotic dynamics in complex high-dimensional control systems, we now investigate the impact this has on the reward. In the MDP framework, the reward function $R : \mathbb{S} \times \mathbb{A} \to \mathcal{R}$ provides feedback to the agent describing how optimal the current state and action are. Using this feedback, the policy is trained to maximise the expected sum of rewards $\sum_{t=0}^{\infty} \gamma^t r_t$ for a given discount factor $\gamma \in [0, 1]$. Moreover, as the action taken is a non-linear transformation of the current state via the control policy $\pi$, the reward function can be redefined as $R(\boldsymbol{s}, \pi(\boldsymbol{s}; \phi))$. Therefore, for a fixed control policy, this function acts as a deterministic non-linear mapping from state space to a one-dimensional reward space in which the stability of the reward can be measured.

By considering the reward over a state trajectory as a trajectory in a one-dimensional reward space, the stability of the reward function can be measured using the Lyapunov Exponents. However, as this space only contains one dimension the SLE cannot be calculated as this value is dependent on the existence of multiple Lyapunov Exponents. Despite this, the MLE can still be used to reliably identify the types of reward dynamics as a negative value indicates stable trajectories with small perturbations to the state of the system producing similar long-term rewards. Moreover, as the reward function is bounded, the reward trajectories cannot diverge indefinitely thus, a positive MLE indicates long-term reward is chaotic and highly sensitive to small changes in the system's state.

Calculating the reward MLE for each policy-environment interaction (Figure 5) shows that reward trajectories in simple low-dimensional environments (*Pointmass*, *Cartpole*) are stable. This is consistent with the state MLE and reward function for these environments as a large reward is given for reaching a single fixed point. Conversely, for high-dimensional systems, the reward MLE is positive indicating a high sensitivity to small changes in system state. This instability in reward is a direct result of the chaotic state dynamics produced by the control system as small changes in system state produce differing state trajectories which attain distinct rewards. However, in the case of the *Walker Stand* task, there is a possibility the true reward MLE is zero or negative for the *SAC* and

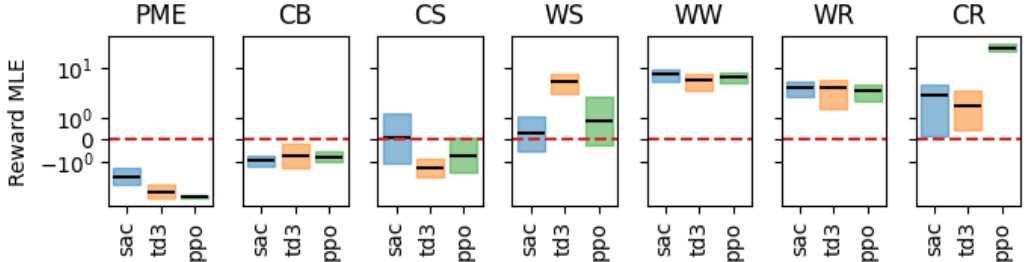

Figure 5: Average reward MLE for the *Pointmass Easy* (PME), *Cartpole Balance* (CB), *Cartpole Swingup* (CS), *Walker Stand* (SW), *Walker Walk* (WW), *Walker Run* (WR) and *Cheetah Run* (CR) when controlled by SAC, TD3 and PPO. Each policy-environment combination is independently trained with 10 random seeds and the average metrics are calculated using 8 initial states.

*PPO* policies. This means that the state trajectories in these systems attain similar rewards despite exponentially diverging. This is consistent with the definition of reward function for this task as it only constrains two dimensions (Torso Height, Torso Angle) to specified values while allowing the remaining 16 dimensions to take any value. However, despite attaining stable rewards, this control system does not produce stable dynamics limiting its application to real-world problems.

## 6 IMPLICATIONS OF THE REWARD FUNCTION

Having established the presence of chaotic state and reward dynamics produced by the policy-environment interaction we now aim to improve the stability of these solutions through reward function modification. In particular, we address the instability of the state space introduced by an unconstrained reward function for environments with static solutions. In these tasks, a high reward is provided for attaining a specified position in a subset of dimensions while allowing the remaining dimensions to take any value. This mismatch between state dimension and constrain dimension allows for a wide range of solutions to be produced which all attain good performance despite having different dynamics. However, this flexibility allows a subset of state dimensions to be ignored during the optimisation process leading to chaotic dynamics.

To investigate the effect the dimensionality of the reward function has on the types of dynamics produced by the MDP we introduce a new control environment *Fixed Point Walker Stand*. This uses the same transition dynamics as the standard *Walker Stand* task however the reward function is conditioned on all dimensions of the state space rather than just torso height and angle. For each state, the reward for *FP Walker Stand* is given by

$$r(\boldsymbol{s}) \ = \ \sigma(-||\bar{\boldsymbol{s}} - \boldsymbol{s}||), \tag{3}$$

where $\sigma$ is the sigmoid function and $||.||$ is euclidean distance between current state $\boldsymbol{s}$ and desired state $\bar{\boldsymbol{s}}$. To identify a desired state, $\bar{\boldsymbol{s}}$, we train an instance of TD3 on the standard *Walker Stand* environment and identify a high reward trajectory with no variation in system states after 700 environment steps. As there is little variation but high sensitivity to initial conditions we can determine this is an unstable fixed point in the dynamical system. Therefore, updating the reward function to minimise the distance between all system states and the fixed point should produce a policy which is robust to changes in initial conditions.

By modifying the reward function in this way this new environment can be compared with the standard *Walker Stand* task to identify if the reward function or transition function produces chaotic dynamics. To this end, we independently train 10 instances of SAC, TD3 and PPO on the standard *Walker Walk* task for 5 Million (10 Million for PPO) environment steps which can provide a baseline for performance and stability. Each policy is then trained for a further 1 Million environment steps using the modified reward function in Eq. 3 before being evaluated. Figures 6a, 6c provides the MLE and SLE for the state trajectories for the *Walker Walk* and *Fixed Point Walker Walk* tasks and shows

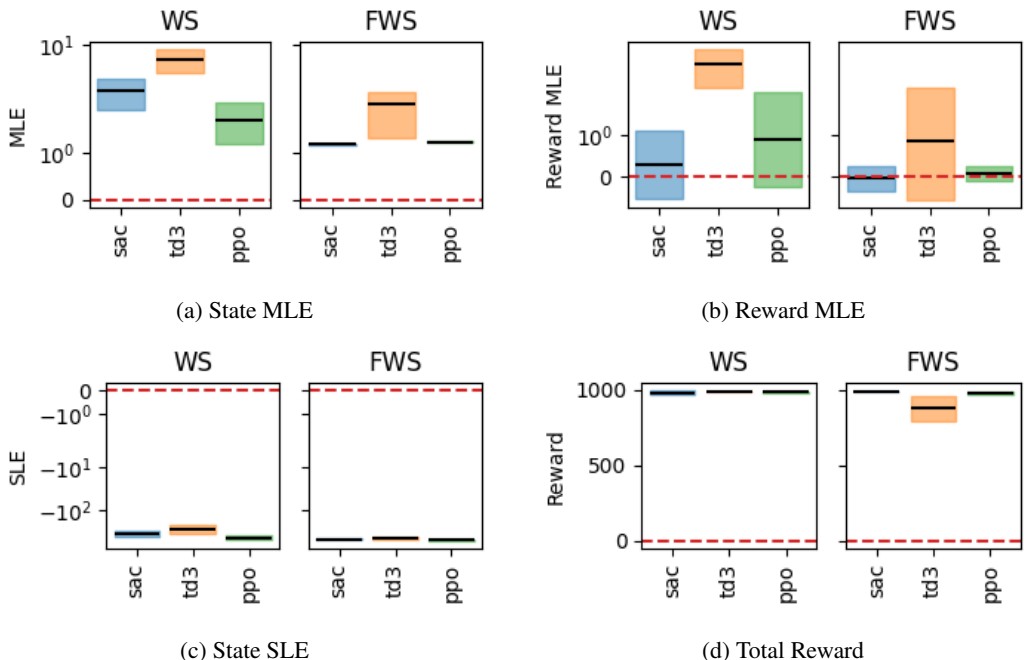

Figure 6: Average state MLE (a), SLE (c), average reward MLE (b) and total reward (d) for the *Walker Stand* (SW) and *Fixed Point Walker Stand* (FWS) when controlled by SAC, TD3 and PPO. Each policy-environment combination is independently trained with 10 random seeds and the average metrics are calculated using 8 initial states.

the constrained task produces less chaotic state dynamics. Moreover, the reward MLE (Figure 6b) also decreases indicating the solutions produced are much more robust to small perturbations in the systems state. However, this increase in stability comes at the cost of performance as the total reward (Figure 6d) for each policy has decreased. Therefore, including a constrained reward function in the MDP introduces a trade-off between flexibility and stability of solutions.

## 7 CONCLUSION

Despite recent progress in Deep Reinforcement Learning, there have been relatively few applications in real-world domains as this method lacks theoretical stability guarantees. In this work, we set out to identify the stability of RL controllers for continuous control environments by viewing the policy-environment interaction as trajectories in a controllable dynamical system. Through this redefinition, we established that when any RL method controls a complex environment the interaction is highly sensitive to initial conditions. As a result, any small changes in the system state compound to produce significantly different long-term outcomes, a sign of unstable dynamics. Furthermore, we demonstrate this level of chaos remains consistent through training despite exploring more of the state space as the policy is only optimised to maximise reward. Finally, we propose a novel method for improving the stability of RL policies for continuous control environments via reward function modification. We show that when the reward function constrains all dimensions of the system state, the MLE of the interaction significantly reduces producing stable dynamics. This demonstrates instability in the MDP arises from the reward function as an unconstrained reward function allows for the existence of multiple distinct solutions from a single policy.

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
