# OpenReview forum: "Investigating the chaotic dynamics produced by deep reinforcement learning controllers"
_ICLR.cc/2024/Conference — Submitted to ICLR 2024_

### Official Review · Reviewer_zKDP · 2023-10-24

**Soundness:** 1 poor
**Presentation:** 2 fair
**Contribution:** 2 fair
**Rating:** 3
**Confidence:** 4

**Summary:**

This paper established that small changes in the system state compound to produce significantly different long-term outcomes for RL algorithms and proposed a novel method for improving the stability of RL policies for continuous control environments via reward function modification.

**Strengths:**

This simulation in this paper illustrates that small initial inaccuracies in sensor readings or actuator movements compounding over time can produce significantly different long-term outcomes

**Weaknesses:**

1. The paper introduces several unfamiliar terms such as "TRAJECTORY STABILITY," "Sensitivity," and "Chaotic," without providing sufficient context or clarification. It is imperative for the author to offer clear definitions of these terms and elaborate on how they relate to Lyapunov stability, which readers might be more acquainted with.

2. The paper repeatedly emphasizes that the MDP framework can be reinterpreted as a controllable dynamical system. However, this assertion is widely acknowledged and lacks the novelty; therefore, continually highlighting it does not contribute significantly to the paper's academic value.

3. The paper is marred by poor writing and grammatical errors, an issue that is evident in sentences like, "One key property of a trajectory which a paramount to dynamical systems theory is it’s stability." Additionally, the results presented in Figure 5 are convoluted, suggesting that the author needs to refine their writing and data presentation skills.

4. The proposed method of "constraining all system states in the reward function" is perplexing. The paper fails to articulate the intuition and logic behind this approach, making it difficult for readers to comprehend its purpose and effectiveness.

5. There is a conspicuous absence of discussion concerning the existing body of research on the stability of reinforcement learning (RL) policies. The author neglects to reference any prior studies in this area, indicating a need for more comprehensive literature review and research on the topic.

**Questions:**

No

---

> ### Author Response · Authors · 2023-11-22
>
> We would like to express our gratitude for taking the time to review our paper and providing valuable feedback. Your comments have been insightful and have certainly contributed to the refinement of our work. We appreciate the effort you have put into the review process. After carefully considering your comments, we would like to address the concerns and criticisms raised during the review.
>
> - **The paper introduces several unfamiliar terms such as "TRAJECTORY STABILITY," "Sensitivity," and "Chaotic," without providing sufficient context or clarification.**
> In Section 2 of the revised version, we provide a more detailed explanation of each term.
>
> - **The paper repeatedly emphasizes that the MDP framework can be reinterpreted as a controllable dynamical system.**
> This was included to provide clarity in case the reader was unfamiliar with the relation between the MDP and dynamical systems. This definition is fundamental for measuring the stability of the control system so has been included to provide clarity.
>
> - **The paper is marred by poor writing and grammatical errors.**
> This is something we have taken a close look at and have attempted to improve in the latest draft.
>
> - **Additionally, the results presented in Figure 5 are convoluted**
> We appreciate this figure was unclear in the original text and have replaced it with a plot showing the mean and confidence interval. We have also extended this to the remaining environment to provide a more robust result.
>
> - **The proposed method of "constraining all system states in the reward function" is perplexing.**
> Our motivation for the constrained reward function was perhaps unclear in the original draft but this is something we have worked on for the revision. In the latest draft, we introduce a method for calculating the MLE of the reward over a trajectory and use this to identify if the reward function is stable, unstable or chaotic. We discover that constrained reward functions produce stable rewards with negative reward MLE however these results are limited to low-dimensional environments. The method of constraining the reward function for the Walker Stand task is then used to further demonstrate that a constrained reward function produces stable dynamics but now in high-dimensional systems.
>
> We hope these revisions adequately address your concerns, and we believe they have strengthened the overall quality of our paper. We appreciate your thorough evaluation and constructive feedback, as it has contributed to the improvement of our research.
>
> Thank you once again for your time and consideration.

---

### Official Review · Reviewer_Hboh · 2023-10-25

**Soundness:** 2 fair
**Presentation:** 3 good
**Contribution:** 2 fair
**Rating:** 3
**Confidence:** 4

**Summary:**

This work investigates the chaotic dynamics of some continuous control tasks with the policy produced by deep reinforcement learning methods. By calculating the maximal Lyapunov exponent (MLE) numerically, they show that simple environments with low dimensions are more robust with respect to the small perturbations to the initial conditions, while complex environments with high dimensions are prone to produce unstable chaotic dynamics. To solve this issue, the authors propose to redesign the reward function such that it counts all the system states. Experiment study shows that such modification reduces the MLE but sacrifices the agent performance.

**Strengths:**

This paper is well-presented and easy to follow. The motivation for this work is strong. The stability of DRL controllers is indeed an important topic, especially when we want to apply such learning-based controllers to real-time systems.

This paper contributes to this topic by investigating the chaotic dynamics produced by DRL controllers, and through several numerical studies, they show that learned controllers are indeed sensitive to the small perturbations of initial conditions. In addition, they propose to redesign the reward function to mitigate such issues.

**Weaknesses:**

The authors quantify the instability of the system by using the sensitivity of a dynamical system to the initial condition. However, a sensitive controller does not necessarily mean unstable from a control performance perspective. For example, Fig 1 shows sensitive dynamics, but it is hardly to be recognized as unstable. In addition, under the definition of sensitivity in this paper (Definition 1), a system seems very easy to be sensitive to since $\beta$ is only assumed to be positive (can be arbitrarily close to zero), and they only require there exist a time step $k$ such that $\| ^k u(s) - ^k u(\hat{s}) \| > \beta$.

The modification of the reward function (2) is weak to me. In order to obtain reference value $\bar{s}$, one has to run the standard problem first, this is rather inefficient. I believe such a method is not practical.

**Questions:**

1. In Fig 5, how to compute the MLE during the training phase?
2. The MLE of the walker walk problem shown in Fig 5 seems inconsistent with the ones in Table 2, are they different sets of experiments?

I would also appreciate the authors to comment on my concerns in the Weakness section.

---

> ### Author Response · Authors · 2023-11-22
>
> We would like to express our gratitude for taking the time to review our paper and providing valuable feedback. Your comments have been insightful and have certainly contributed to the refinement of our work. We appreciate the effort you have put into the review process. After carefully considering your comments, we would like to address the concerns and criticisms raised during the review.
>
> - **A sensitive controller does not necessarily mean unstable from a control performance perspective.**
> Our experiments show that instability in RL policies also causes instability in the rewards of those policies. To clarify this in the manuscript, we introduce a method for calculating the MLE of the reward over a trajectory and use this to identify if the reward function is stable, unstable or chaotic. We show that for simple low-dimensional environments, reward is stable with exponentially converging reward trajectories. Conversely, for high-dimensional environments, this reward is highly sensitive to initial conditions. As a result, policies which produce unstable state dynamics also produce unstable reward trajectories.
>
> - **The modification of the reward function (2) is weak to me.  In order to obtain reference value, one has to run the standard problem first, this is rather inefficient.**
> This is perhaps something which we did not make clear in the original draft. In section 5 each agent is initially trained for 5 Million (SAC, TD3) or 10 Million (PPO) environment steps. This allows each policy to find a solution to the task and optimize this to attain a high reward. Having completed this training a reference value can be calculated for each policy and used for the constrained reward function. Each policy is then trained for a further 1 Million environment steps using this updated policy in order to improve the stability of the solution. In this sense, the proposed approach can be seen as fine-tuning the policy for stability
>
> - **In Fig 5, how to compute the MLE during the training phase?**
> During training a frozen copy of the control policy is combined with the control environment to produce a single, deterministic function which maps states to states. Using this update rule the spectrum of Lyapunov exponents is then calculated using the method outlined by Benettin et al and the MLE is taken as the largest value in the spectrum of Lyapunov Exponents. We have clarified this in Section 4.
>
> - **The MLE of the walker walk problem shown in Fig 5 seems inconsistent with the ones in Table 2, are they different sets of experiments?**
> This inconsistency has been updated to reflect the results from Section 4.
>
> We hope these revisions adequately address your concerns, and we believe they have strengthened the overall quality of our paper. We appreciate your thorough evaluation and constructive feedback, as it has contributed to the improvement of our research.
>
> Thank you once again for your time and consideration.

---

### Official Review · Reviewer_rHqm · 2023-10-27

**Soundness:** 3 good
**Presentation:** 3 good
**Contribution:** 2 fair
**Rating:** 6
**Confidence:** 3

**Summary:**

The paper explored the dynamics of controllers that have been trained using various reinforcement learning techniques. It analyses the MDP solutions by attempting to quantify their stability. The authors' analyses suggest that these tend to be chaotic for complex environments and agents. The work concludes with a practical guideline to mitigate these instabilities using modified reward functions.

**Strengths:**

The paper is well written, with a clear structure and suitable diagrams.

A main strength of the paper is the interesting problem that it tackles, which represents solution stability, and is part of a wider field dedicated to understanding and perfecting the process of embodying real-life agents, using policies trained in virtual environment.

Another strength of the work is the analysis of a range of RL algorithms, suggesting that the problem the author's identified is present in many systems.

**Weaknesses:**

A weakness of the paper is that only one environment is used, the Walker Walk, to show that a modified reward function can increase the stability of the solutions.

Another weakness is that while a range of RL algorithms are considered, they are not as varied as they could be. The authors should consider other algorithms types and show their dynamics. For example dynamics in off vs on-policy algorithms, or value vs policy based algorithms.

**Questions:**

1. What would be the results of section 5 if the walker stand and run would be considered?
2. Please define the co-domain A of the function/policy \pi in section 3.
3. What are the precise states s that were used to compute the dynamics characteristics in the experiments?
4. From figure 2, it seems like PPO has difficulties solving CR. Is this why in figure 4 we see PPO having an unusually high MLE for CR? If so, this needs to be addressed in the paper.
5. Please expand figure 5 to include the other tasks as well.

---

> ### Author Response · Authors · 2023-11-22
>
> We would like to express our gratitude for taking the time to review our paper and providing valuable feedback. Your comments have been insightful and have certainly contributed to the refinement of our work. We appreciate the effort you have put into the review process. After carefully considering your comments, we would like to address the concerns and criticisms raised during the review.
>
> - **The authors should consider other algorithm types and show their dynamics.**
> This is something we would like to achieve but was not finalized at the time of submission. However, note that the selection of algorithms tested covers both on- and off-policy. In Section 4 we have clarified the definition of each model and specified if they are On Policy (SAC, TD3) or Off Policy (PPO).
>
> - **Please define the co-domain A of the function/policy \pi in section 3.**
> This definition has been added to Section 3.
>
> - **What are the precise states s that were used to compute the dynamics characteristics in the experiments?**
> The states used to calculate the Lyapunov Exponents are taken from sample trajectories with minimum length 200 and maximum length 800. This allows the trajectories to reach any chaotic attractors while still remaining in their training data distribution. We have updated section 4 to clarify this point.
>
> - **From figure 2, it seems like PPO has difficulties solving CR. Is this why in figure 4 we see PPO having an unusually high MLE for CR?**
> While the lower performance of PPO does have an impact on the dynamics produced by the Cheetah Run environment, we have also found that the sum of all the Lyapunov exponents is positive which indicates diverging dynamics. As a result, the trajectories for this system are not bound by a chaotic attractor and can diverge indefinitely in the phase space. This lack of converging dynamics allows for a larger MLE.
>
> - **Please expand figure 5 to include the other tasks as well.**
> This figure has been updated to include the other environments. Furthermore, the raw values have been replaced with the mean and standard deviation at each time step to provide clarity.
>
> We hope these revisions adequately address your concerns, and we believe they have strengthened the overall quality of our paper. We appreciate your thorough evaluation and constructive feedback, as it has contributed to the improvement of our research.
>
> Thank you once again for your time and consideration.

---

> > ### Comment · Reviewer_rHqm · 2023-11-23
> > **Thank you**
> >
> > Thank you to the authors for providing clarifications and edits. I very much enjoyed this particular piece of research.

---

### Official Review · Reviewer_y5dK · 2023-10-31

**Soundness:** 1 poor
**Presentation:** 3 good
**Contribution:** 1 poor
**Rating:** 3
**Confidence:** 3

**Summary:**

Paper proposes a technique for measuring the stability of reinforcement learning policies, demonstrates that standard algorithms tend to be unstable for high dimensional environments, and proposes a modification to the reward function to improve the stability of RL policies.

**Strengths:**

Significance
- Paper studies an interesting and important problem -- RL policies tend to be sensitive to initial conditions.

Clarity
- Figures and tables look nice, paper is written nicely and easy to understand.

**Weaknesses:**

It seems like the main contribution of the paper is in identifying a reason for policy instability (only a subset of the system state is used in the reward) and proposing a solution to enable more stable policies (constraining all system states in the reward function). However, *it is not clear whether stability, in the way it is defined in the paper, is actually a desirable characteristic of RL policies*.

As a concrete example, let us consider walker stand. In the default version of the environment, the system state include features such as the agent's angle of the foot joint. which is not used the the reward function (reward is only dependent on torso height and angle). As a result, RL policies, given different initial conditions, may control the agent to position its foot joint in different angles while maintaining a standing position. This behavior will lead to consistently high reward across different initial conditions, while yielding a low stability score (as defined by the Maximal Lyapunov Exponent), because the stability score will punish the policy for putting the agent's foot at different angles in different trajectories. Moreover, the paper's proposed method of modifying the reward function will minimize this behavior by incentivizing the policy to always put the agent's foot in the same angle, which will lead to a higher stability score. However, it is not clear that this "stable" behavior is actually desirable. In the example of walker stand, the angle of the foot joint is not part of the reward function, because it *doesn't matter* for the performance of the task. In the current results in the paper, improving stability via the modified reward function does not consistently lead to better performance (in terms of the reward), so it is not clear why measuring and improving stability is meaningful or significant.

**Questions:**

Is there a scenario where improving the stability of a policy will improve the return of the policy, e.g. by providing a more shaped reward signal?

---

> ### Author Response · Authors · 2023-11-22
>
> We would like to express our gratitude for taking the time to review our paper and providing valuable feedback. Your comments have been insightful and have certainly contributed to the refinement of our work. We appreciate the effort you have put into the review process. After carefully considering your comments, we would like to address the concerns and criticisms raised during the review.
>
> -  **It is not clear whether stability is actually a desirable characteristic of RL policies.**
> This is a fair question. Our experiments show that instability in RL policies also causes instability in the rewards of those policies. To clarify this in the manuscript, we introduce a method for calculating the MLE of the reward over a trajectory and use this to identify if the reward function is stable, unstable or chaotic. We show that for simple low-dimensional environments, reward is stable with exponentially converging reward trajectories. Conversely, for high-dimensional environments, this reward is highly sensitive to initial conditions. As a result, policies which produce unstable state dynamics also produce unstable reward trajectories.
>
> We hope these revisions adequately address your concerns, and we believe they have strengthened the overall quality of our paper. We appreciate your thorough evaluation and constructive feedback, as it has contributed to the improvement of our research.
>
> Thank you once again for your time and consideration.

---

### Meta-Review · Area_Chair_EQZK · 2023-12-24

**Metareview:**

### Summary

- proposed a novel method for improving the stability of RL policies for continuous control environments via reward function modification.
- Proposes to compute Maximal Lyapunov exponent (MLE) numerically, and highlights that complex environments with high dimensions are prone to produce unstable chaotic dynamics.


###  Strengths
+ studies the problem formal robustness in RL
+ proposes a solution to fine-tune models to improve robustnes

### Weaknesses:
- Lack of clarity and self-containments in writing, especially since the paper addresses the intersection control theory and ML community.
- Assumes that the pretrained policy indeed achieves optimum for reward modificatio.
- Lack of thorough evaluation in types of perturbation: noise in dynamics, action space, observation space
- Comparison to robustness through randomization.

**Justification For Why Not Higher Score:**

The current ratings are supported by multiple reviews, and the AC does not find evidence otherwise to overturn.

**Justification For Why Not Lower Score:**

The paper presents an interesting new setup for robustness of RL for control and presents an interesting set of experiments, albeit on rather small and unrealistic benchmarks.

---

### Decision · Program_Chairs · 2024-01-16

Reject